# Characterization of time-variant and time-invariant assessment of suicidality on Reddit using C-SSRS

Manas Gaur[1]*, Vamsi Aribandi[2], Amanuel Alambo[2], Ugur Kursuncu [1], Krishnaprasad Thirunarayan[2], Jonathan Beich[3], Jyotishman Pathak[4], Amit Sheth[2]

**1** Artificial Intelligence Institute, University of South Carolina, Columbia, SC, United States of America, **2** Kno. e.sis Center, Wright State University, Dayton, OH, United States of America, **3** Department of Psychiatry, Wright State University, Dayton, OH, United States of America, **4** Department of Population Health Sciences, Weill Cornell Medicine, New York, NY, United States of America

* mgaur@email.sc.edu

## Abstract

Suicide is the 10$^{th}$ leading cause of death in the U.S (1999-2019). However, predicting when someone will attempt suicide has been nearly impossible. In the modern world, many individuals suffering from mental illness seek emotional support and advice on well-known and easily-accessible social media platforms such as Reddit. While prior artificial intelligence research has demonstrated the ability to extract valuable information from social media on suicidal thoughts and behaviors, these efforts have not considered both severity and temporality of risk. The insights made possible by access to such data have enormous clinical potential—most dramatically envisioned as a trigger to employ timely and targeted interventions (i.e., voluntary and involuntary psychiatric hospitalization) to save lives. In this work, we address this knowledge gap by developing deep learning algorithms to assess suicide risk in terms of severity and temporality from Reddit data based on the Columbia Suicide Severity Rating Scale (C-SSRS). In particular, we employ two deep learning approaches: time-variant and time-invariant modeling, for user-level suicide risk assessment, and evaluate their performance against a clinician-adjudicated gold standard Reddit corpus annotated based on the C-SSRS. Our results suggest that the time-variant approach outperforms the time-invariant method in the assessment of suicide-related ideations and supportive behaviors (AUC:0.78), while the time-invariant model performed better in predicting suicide-related behaviors and suicide attempt (AUC:0.64). The proposed approach can be integrated with clinical diagnostic interviews for improving suicide risk assessments.

## Introduction

Social media provides an unobtrusive platform for individuals suffering from mental health disorders to anonymously share their inner thoughts and feelings without fear of stigma [1]. Seventy-one percent of psychiatric patients, including adolescent, are active on social media [2, 3]. Suicide is an often-discussed topic among these social media users [4]. Further, it may

**Data Availability Statement:** The data is available through the public repository Zenodo at https://doi.org/10.5281/zenodo.4543776.

**Funding:** Amit Sheth, Jyotishman Pathak, Krishnaprasad Thirunarayan, 1 R01 MH105384-01A1, National Institute of Mental Health, https://federalreporter.nih.gov/Projects/Details/?projectId=891050 Amit Sheth, Krishnaprasad Thirunarayan, 5R01DA039454-02, National Institute on Drug Abuse, https://projectreporter.nih.gov/project_info_description.cfm?projectnumber=5R01DA039454-02 Amit Sheth, CNS-1513721 National Science Foundation, https://www.nsf.gov/awardsearch/showAward?AWD_ID=1513721 Amit Sheth, NSF Award 1761931, Spokes: MEDIUM: MIDWEST: Collaborative: Community-Driven Data Engineering for Substance Abuse Prevention in the Rural Midwest. The funders had no role in study design, data collection and analysis, decision to publish, or preparation of the manuscript.

**Competing interests:** The authors have declared that no competing interests exist.

serve as an alternative resource for self-help when therapeutic pessimism exists between the mental healthcare providers (MHPs) and patients [5–8]. Leavey et al. discuss the challenges of cultural and structural issues (e.g., the social stigma of a depression diagnosis), limited provider-patient contact time, over-reliance on medication use, inadequate training to fully appreciate the nuance of a multifactorial disease, lack of access to mental health services, and a sense of mistrust in broaching the topic of suicide [9]. These factors may coalesce into fragmented patient care due to frequently switching providers or fleeing psychiatric care altogether with possible consequences of deteriorating conditions leading to a suicide attempt [10, 11].

Therapeutic pessimism concerns with the widely held belief that psychiatric patients are extremely difficult to treat, if not immune to treatment [12]. In this study, we focus on the terminal mental health condition, suicide and investigate methods to estimate suicide risk levels. The progression from suicide-related behaviors and ultimately a suicide attempt is gradual and often follows a sinusoidal trajectory of thought and behavior severity through time [13]. Such drift in suicide risk level can be attributed to personal factors (e.g., loss of someone close), external events (e.g., pandemic, job loss), concomitant mental health disorders (depression, mania, psychosis, and substance intoxication and withdrawal), or comorbid chronic health problems [14]. The time-variant changes in these factors make the task of diagnosis for MHPs more challenging [15]. Even though Electronic Health Records (EHRs) are longitudinal, studies have predominantly relied on *time-invariant modeling* of content to predict suicide-related ideations, suicide-related behaviors, and suicide attempt [16, 17]. This approach is often employed due to patients' low engagement and poor treatment adherence resulting ill-informed follow-up diagnostic procedure.

Time-invariant modeling is the aggregation of observational data from a patient's past visits to estimate the severity of mental illness and suicide risk. One practical approach is to monitor alternative sources, such as Reddit posts, over a specified time period to detect time-variant language drifts for signals of suicide-related ideations and behaviors. Signal detected in this information would complement in patient's understanding from the EHR data. In a comparison of Reddit and EHR data in the NYC Clinical Data Research Network (CDRN), it was observed that patients communicated regularly on topics of self-harm, substance abuse, domestic violence, and bullying—all of which are suicide risk factors [18, 19].

Complementary to the above effort, suicide-related behavioral assessment using *time-variant modeling* over social media platforms has been promising [20]. Time-variant modeling involves extracting suicide risk-related information independently from a sequence of posts made by a user. Such an approach allows you to capture the explainable patterns in suicide-related ideations, suicide-related behaviors, and suicide attempts, similar to the process of MHPs identifying these risk levels.

Multidisciplinary research teams involving computer scientists, social scientists, and MHPs have recently increased emphasis on understanding timestamped online users' content obtained from different social communication platforms such as Twitter, Reddit, Tumblr, Instagram, and Facebook [21–26]. Of these platforms, patients reported that Reddit was the most beneficial option in helping them cope with mental health disorders because of the pre-categorized mental health-related subreddits that provide an effective support structure.

Reddit is one of the largest social media platforms with >430 million subscribers and 21 billion average screen visits per month across >130,000 subreddits. On per month average, around 1.3 million subscribers anonymously post mental health-related content in 15 of the most active subreddits pertaining to mental health (MH) disorders ($\sim$42K post on r/Suicide-Watch) [27]. The analysis of Reddit content is demanding due to a number of reasons, including interaction context, language variation, and the technical determination of clinical relevance. Correspondingly, potential rewards of greater insight into mental illness are in

general and suicidal thoughts and behavior specifically is great. These broader observations of challenges translate into three aspects of modeling and analysis: (1) Determination of *User-Types*, (2) Determination of *Content-Types*, and (3) *Clinical grounding.*

There are three *User-Types* in mental health subreddits (MH-Reddits):

1. *Non-throwaway accounts*: support seeking users who have mental illness and are not affected by social stigma.

2. *Throwaway accounts*: support seeking users who anonymously discuss their mental illness by creating accounts with username containing the term "throwaway" [28].

3. *Support Providers/Supportive*: users who share their resilience experiences.

*Content-Types* on MH-Reddits capture (1) ambiguous and (2) transient postings made by different users in User-types.

1. *Ambiguous content-type* include postings from users who are not currently struggling with their own suicidal thoughts and behaviors but might have received treatment (e.g., targeted psychotherapy modalities, psychoactive medications) and become "supportive" by sharing their helpful experiences. The content of supportive users is ambiguous compared to suicidal users because they typically describe their struggles, experiences, and coping mechanisms. This content's bimodal nature poses significant challenges in information processing and typically leads to higher false positives.

2. *Transient posting* refers to the content published by a single user in multiple different MH-Reddit communities. Analyzing transient posts made by such a user using both time-variant and time-invariant techniques provides a more comprehensive evaluation of user intentions with a potentially higher clinical value. These transient phases provide meaningful signals about the intensity, frequency/persistence, and impact of MH disorder symptoms on quality of life. For example, a user makes a post on r/BPD (a subreddit dedicated to the topic of borderline personality disorder) about obsessiveness, intrusive thoughts, and failure to cope with sexual orientation issues. A week later, the same user post's in r/Suicide-Watch about isolation, sexual assault, ignorant family members, and worthlessness in a relationship. Later that same day the user makes another post in r/Depression about hopelessness, betrayal, and abandonment, which has caused restless nights and poor sleep. Finally, a couple days later, (s)he makes a post in r/SuicideWatch about his plan of intentionally isolating and going to sleep forever. These time-sensitive signals provide an informative user-level suicidal thoughts and behaviors risk assessment [29, 30].

Attention must be placed on bridging the gap between "how clinicians actually treat patients" (i.e., clinical grounding) and "how patients express themselves on social media (*User Types and Content Types*)" [4]. A recent survey on suicide risk in the context of social media suggests that existing studies on the topic have ignored clinical guidelines in the risk modeling framework and have placed an over-reliance on statistical natural language processing [31]. With the exceptions of recent studies by Gaur et al. [13, 32], Alambo et al. [33], Chancellor et al. [34], and Roy et al. [35], prior efforts have not explored the utility of the Diagnostic Statistical Manual for Mental Health Disorders (DSM-5) and questionnaires (e.g., PHQ-9, CES-D, C-SSRS) in the context of analyzing social media data.

Assessment of an individual's suicide risk level is challenging in social media due to time-invariant, and time-variant patterns in language, sentiment, and emotions in the pursuit of more informed psychiatric care [36–38]. Although these objective markers can screen patients, their utility to MHPs has not been validated. Li et al. utilize language, emotions, and

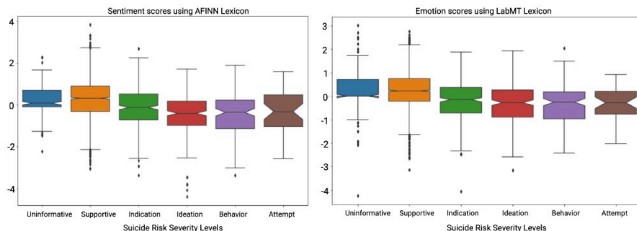

**Fig 1. No significant variation in sentiment (using AFINN) and emotions (using LabMT) across suicide risk severity levels in C-SSRS.**

subjectivity cues to compute the Mental Health Contribution Index (MHCI). The MHCI measures user engagement in MH-Reddits to assess for mental health disorders and their respective symptoms [39]. The authors identified a substantial correlation between linguistic cues (e.g., increased use of verbs, subordinate conjunctions, pronouns, articles, readability) with MHCI and identified them as indicators of increased mental health problems [39]. Further, different user types and content types have varied influences on MHCI, which is different from the influence of linguistic cues. Our experiment with C-SSRS-based labeled dataset revealed that sentiment (Fig 1(left)) and emotion (Fig 1(right)) factors did not satisfactorily discriminate different suicide risk severity levels [13].

The limitations associated with the annotation process, inter-rater agreement, and clinical translation of prior research have been identified as significant concerns [40, 41]. A more substantial concern is that despite using state-of-the-art algorithms (e.g., multi-task learning by Benton et al. [42]), no significant improvement was observed in the estimation of MH severity, particularly suicide risk severity. Various syntactic, lexical, [20, 43] psycholinguistic features using linguistic inquiry and word count (LIWC), [44] key phrases, [45] topic modeling, [46, 47] and a bag of word models [48] or other data-driven features have been explored for modeling the language used in online mental health conversations. However, these methods' clinical relevance has not been investigated [49–51].

Our study aims to fill these gaps of prior studies by taking into account *User Types*, *Content Types*, and an ubiquitous *clinical screening tool*, C-SSRS, to assess suicide risk severity in a time-variant and time-invariant manner [52]. To study user behavior on Reddit with clinical semantics, we made category-level additions to C-SSRS. The novel category of users: "Supportive" and "Suicide Indication" allowed us to examine suicide risk severity levels on Reddit more accurately. Through the support of practicing psychiatrists in performing annotation and achieving substantial inter-rater agreement, we strengthen the clinical relevance of our research and inferences derived from outcomes [53].

Our study is also structured to supplement a patient's EHR by providing community-level markers with the added benefit of allowing MHPs to monitor patient activity on social media. While some mental health patients are fortunate enough to see multiple members of a care team on a frequent (weekly at best) basis, and most are not seen for months at a time. The potential changes in the patients' state-of-mind during such an extended time period, due to internal or external factors, could have catastrophic results. Due to existing and increasing work demands on MHPs to care for increasing numbers of patients in the same or less amount of workday time, any proposal to add the responsibility of screening patients' social media posts for concerning content is not realistic. Any effort to shift this load to human labor would be prohibitively expensive. Our study can also be used to build tools that quantify suicide risk based on mental health records (e.g., clinic notes, psychotherapy notes) in combination with social media posts, resulting in an improved ability for clinical mental healthcare workers (e.g.,

**Fig 2. An overall pipeline of our comparative study to predict suicide-risk of an individual using the C-SSRS and Diagnositic Statistical Manual for Mental Health Disorders (DSM-5).** S: Supportive, I: Suicide Ideation, B: Suicide Behavior, A: Suicide Attempt users.

psychiatrists, clinical psychologists, therapists) and policymakers to make informed patient-centered decisions [54, 55]. The benefit this technology brings to bear on the existing standard-of-care includes more frequent patient monitoring and the ability to find the "needle in the haystack" of voluminous data [56].

We outline our approach in Fig 2. We make the following key contributions in this research: (a) We create a new Reddit dataset of 448 users with both user-level and post-level annotations following C-SSRS guidelines. (b) Through various content-types and user-types illustrated in the mental health community on Reddit, we analyze the expression of suicide-related ideation, suicide-related behaviors, and suicide attempts using medical knowledge sources and questionnaires. (c) The sequential (Time-variant) and convolutional (Time-invariant) deep learning models developed in this study measure the performance and limitations in suicide risk assessment. (d) We describe a clinically grounded qualitative and quantitative evaluation using C-SSRS and show the benefits of a hybrid model for early intervention in suicide risk.

## Materials and methods

We leveraged a dataset comprising of 92,700 Reddit users and 587,466 posts from the r/SuicideWatch subreddit. The dataset spans 11 years from 2005 to 2016, with users transitioning from different mental health subreddits: r/bipolar (BPR), r/BPD (Borderline Personality Disorder), r/depression (DPR), r/anxiety (ANX), r/opiates (OPT), r/OpiatesRecovery (OPR), r/selfharm (SLF), r/StopSelfHarm (SSH), r/BipolarSOs (BPS), r/addiction (ADD), r/schizophrenia (SCZ), r/autism (AUT), r/aspergers (ASP), r/cripplingalcoholism (CRP), and r/BipolarReddit (BPL).

Considering the human behavior in their online written conversation, we followed a Zipf-Mandelbrot law and negation resolution method to computationally identify potential suicidal users (For a detailed description of this method, we refer the reader to Gaur et al. [13] and Fig 3). Furthermore, we utilize domain-specific medical information source and knowledge graphs (DSM-5, DataMed, SNOMED-CT, ICD-10,and Drug Abuse Ontology) to perform MedNorm for normalizing linguistic variations,a challenge in online health communications [57]. Our selection of resources is governed by the structure of the New York City CDRN warehouse, which provides treatment information, condition description, drug exposure, and observation on mental health conditions [58]. Using medical entity normalization (MedNorm), [59] we gathered treatment-related information from SNOMED-CT and ICD-10, [60] observation and drug-related information from Twitter Adverse Drug Reaction (TwADR) and AskaPatient Lexicons, and Drug Abuse Ontology (DAO [61, 62]), and information on Mental

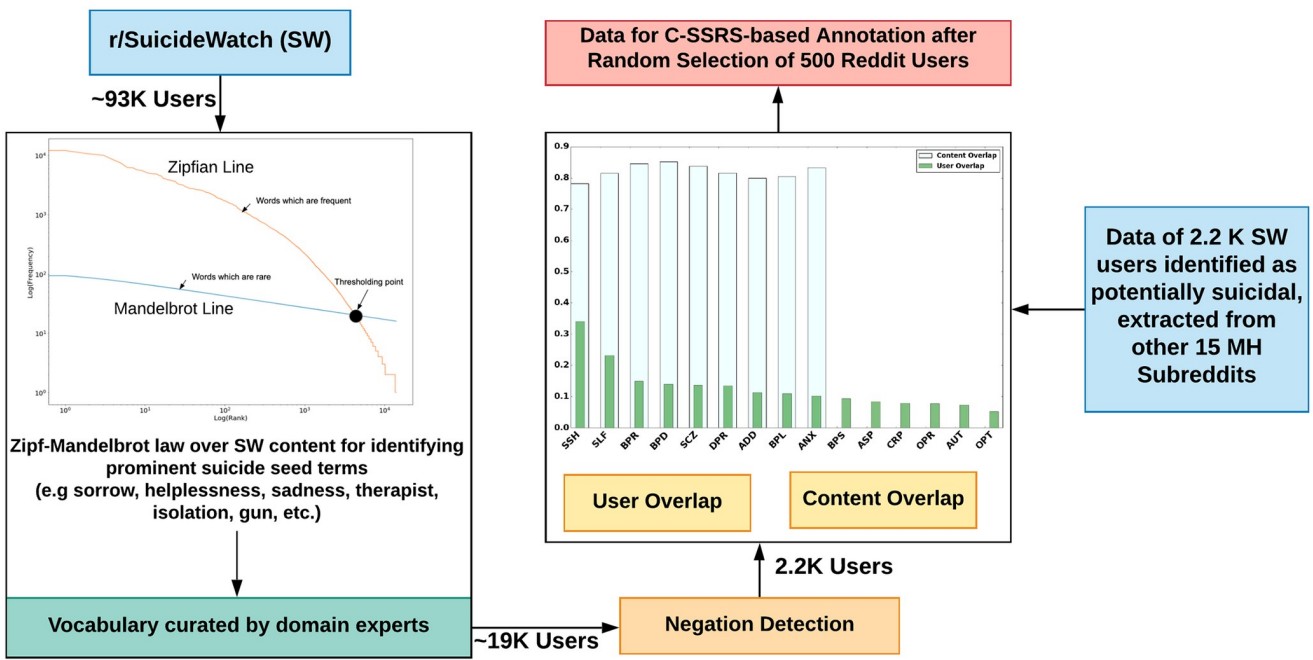

**Fig 3. Procedure for generating the dataset for annotation.** The 500 randomly selected Reddit users were labeled with C-SSRS guidelines and additional labels: Supportive and Suicide Indication. In current research we utilize 448 users (removing 52 suicide indication users) with post-level and user-level annotations.

health conditions from the DSM-5 [63]. This helped us construct a semantics preserving and clinically grounded pipeline that addresses significant concerns overlooked by previous studies on MH systems [5, 64].

For example, consider the following three posts: (P1) *I am sick of loss and need a way out*; (P2) *No way out, I am tired of my losses*; (P3) *Losses, losses, I want to die.* The phrases in P1 and P2 are predictors of suicidal tendencies but are expressed differently, while P3 is explicit [65]. The procedure requires computing the semantic proximity between n-gram phrases and concepts in medical lexicons, which takes into account both syntax and contextual use. Among the different measures for semantic proximity, we utilize cosine similarity measures. The vector representations of the concepts and n-gram phrases are generated using the ConceptNet embedding model [66, 67]. We employ TwADR, and AskaPatient lexicons for normalizing the informal social media communication to their equivalent medical terms [68]. These lexicons are created using drug-related medical knowledge sources and further enriched by mapping twitter phrases to these concepts using convolutional neural networks (CNNs) over millions of tweets [59]. After MedNorm, the post P1 transforms to *I am helpless and hopeless* and post P2 becomes *hopeless, I am helpless*. Thus we obtain syntactically similar normal forms when we have two semantically equivalent posts. To perform MedNorm, we generated a word vector for each word in the post using ConceptNet. We independently computed its cosine similarity with the word vector of concepts in the TwADR and AskaPatient. If the cosine similarity is above an empirical threshold of 0.6, words are replaced with the medical concepts in the lexicons (see Fig 4) [13].

*Privacy or Ethical Concerns*: Our current study performs analysis of community-level posts to examine the utility of social media (e.g., Reddit) as a complementary resource to EHR for informed decision making for mental healthcare. The study does not involve human subjects (e.g., tissue samples), animals (e.g., live vertebrates or higher vertebrates), or organs/tissues for

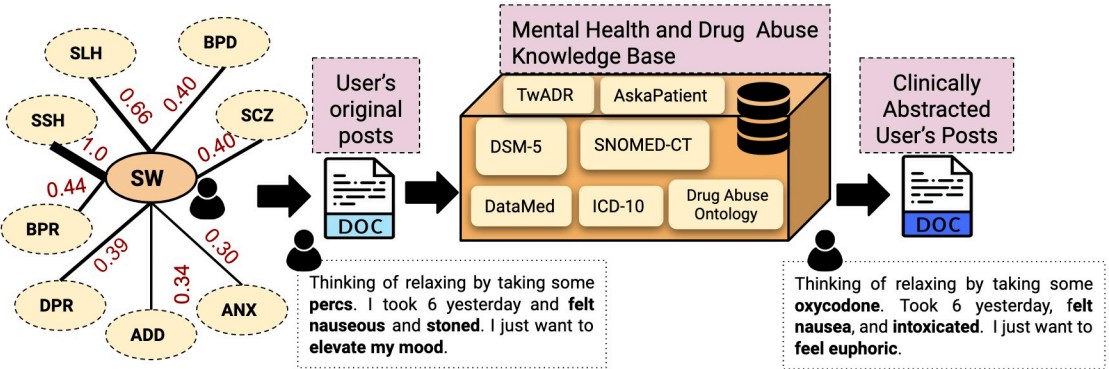

**Fig 4. The transient posting of potential suicidal users in other subreddits requires careful consideration to appropriately predict their suicidality.** Hence, we analyze their content by harnessing their network and bringing their content if it overlaps with other users within r/SuicideWatch (SW). We found, Stop Self Harm (SSH) > Self Harm (SLH) > Bipolar (BPR) > Borderline Personality Disorder (BPD) > Schizophrenia (SCZ) > Depression (DPR) > Addiction (ADD) > Anxiety (ANX) (User overlap between SW and other MH-Reddit is shown by thickness in the connecting line) to be the most active subreddits for suicidal users. After aggregating their content, we performed MedNorm using Lexicons to generate clinically abstracted content for effective assessment.

gathering the data and experiment design. The datasets, to be made public, do not include the screen names of users on Reddit, and strictly abide by the privacy principles of Reddit platform. Example posts provided in this article have been modified for anonymity. Our study has been granted an IRB waiver from the University of South Carolina IRB (application number Pro00094464).

## Suicide risk severity lexicon

Existing studies demonstrate the potential for detecting symptoms of depression by identifying word patterns and topical variations using the PHQ-9 questionnaire, a widely used depression screening tool [27, 69–71]. When suicidal thinking is identified on the PHQ-9, the C-SSRS is used to determine suicide severity [72]. The C-SSRS categorizes suicidal thinking into Suicidal Behaviors, including Ideations and Attempts. Though these categories are well suited for clinical settings, their utilization for user assessment of online social media data is not straightforward. For example: consider a user's post: *Time, I was battling with my suicidal tendencies, anger, never-ending headaches, and guilty conscience, but now, I don't know where they went.* According to C-SSRS, the user exhibits Suicidal Ideation; however, the user is supportive towards a person seeking help. This shows a variation in behaviors exhibited by a user. Hence, in a recent study, we used C-SSRS with Reddit data to assess users' suicidality by augmenting two additional categories to C-SSRS, viz., Supportive and Suicidal Indicator [13]. While the C-SSRS categories are clinically defined, there is a need for a lexicon (with social media phrases and clinical knowledge) that quantifies the relevance of a post to suicidal categories. This study created a suicide severity lexicon with terms taken from social media and medical knowledge sources (SNOMED-CT, ICD-10, Drug Abuse Ontology, and DSM-5 [73]). These medical knowledge resources are stored in graphical structures with nodes representing medical concepts and linked through medical relationships (e.g., TypeOf, CriteriaOf, SubdivisionOf, InclusionCriteria). For example, in SNOMED-CT, *Subject*: {fatigue, loss of energy, insomnia} is associated to *Object*: {Major Depressive Disorder} through *Relation* (or Predicate): {InclusionCriteria}. In ICD-10, Subject: {Major Depressive Disorder with Psychotic Symptoms} is associated with Object: {Suicidal Tendencies} through *Relation* (or Predicate): {InclusionCriteria}. Hence, using these two knowledge sources that contain semantically relevant domain

relationships, one can create a mental health knowledge structure for making inferences on suicidal severity. Following this intuition, the suicide risk severity lexicon for semantically annotating the Reddit dataset was created by Gaur et al. [13].

## Annotator agreement

Our study focuses on comparing two competing strategies, TinvM and TvarM, in assessing an individual's suicidality using their online content for early intervention. To illustrate, we utilize an annotated dataset of 500 Reddit users with the following labels: *Supportive, Suicide Indication, Suicide Ideation, Suicide Behaviors, and Suicide Attempt*. For this research, we removed users labeled with suicide indication, giving a total of 448 users with labels: *Supportive, Suicide Ideation, Suicide Behaviors, and Suicide Attempt* (see Table 1 for statistics on annotated data). Suicide Indication (IN) category separates users using at-risk language from those actively experiencing general or acute symptoms. Users might express a *history of divorce*, *chronic illness*, *death in the family*, or *suicide of a loved one*, which are risk indicators on the C-SSRS, but would do so relating in empathy to users who expressed ideation or behavior, rather than expressing a personal desire for self-harm. In this case, it was deemed appropriate to flag such users as IN because while they expressed known risk factors that could be monitored they would also count as false positives if they were accepted as individuals experiencing active ideation or behavior.

Four practicing psychiatrists have annotated the dataset with a substantial pairwise inter-rater agreement of 0.79, and groupwise agreement of 0.69 [13, 74]. The created dataset allows Time-invariant suicide risk assessment of an individual on Reddit, ignoring time-based ordering of posts. For Time-Variant suicide risk assessment, the posts needed to be ordered concerning time and be independently annotated. Following the annotation process highlighted in Gaur et al. [13] using a modified C-SSRS labeling scheme, the post-level annotation was performed by the same four psychiatrists with an inter-rater agreement of 0.88 (Table 2a) and a group-wise agreement of 0.76 (Table 2b). The annotated dataset of 448 users comprises 1170

**Table 1. Data statistics on the annotated Reddit data for current research and Gaur et al.** The users labeled as *suicide indication* in 500 Reddit user dataset were removed because of high disagreement between annotators during post-level annotation.

| Data Property | Gaur et al. | Current Research without *Suicide Indication* |
|---|---|---|
| Total Number of Users | 500 | 448 |
| Total Number of Posts | 15755 | 7327 |
| Total Number of Sentences | 94083 | 36788 |
| Average Number of Post per User | 31.51 | 18.27 |
| Average Number of Sentences per Post | 6 | 5.02 |
| Types of Annotation | User-Level | User-Level and Post-Level |

**Table 2. Inter-rater reliability agreement using Krippendorff metric.** A,B,C,and D are mental healthcare providers as annotators. The annotations provided by MHP "B" showed the highest pairwise agreement and were used to measure incremental groupwise agreement for the robustness in the annotation task.

| | B | C | D | | A | A&C | A&C&D |
|---|---|---|---|---|---|---|---|
| **A** | 0.82 | 0.79 | 0.80 | | | | |
| **B** | - | 0.85 | **0.88** | **B** | 0.82 | 0.78 | **0.76** |
| **C** | - | - | 0.83 | | | | |
| (a) Pairwise reliability agreement | | | | (b) Groupwise reliability agreement | | | |

supportive (throwaway account: 421, Non-throwaway account: 437) and uninformative (throwaway account: 115, Non-throwaway account: 197) posts. For throwaway accounts, the dataset had 37 supportive users (S), 63 users with suicide ideation (I), 23 users with suicide behavior (B), and 17 users had past experience with suicide attempt (A). User distribution within non-throwaway accounts is as follows: 85 S users, 115 I users, 76 B users, and 33 A users.

## Methods

We explain two competing methodologies: TvarM and TinvM, for suicide risk severity prediction.

**TvarM to suicide risk prediction.** Prior literature has shown the effectiveness of sequential models (e.g., recurrent neural network (RNN), long short term memory (LSTMs)) in learning discriminative feature representations of contextual neighboring words with reference to a target word. Moreover, it has been investigated through experimentation that sentences formed by an individual express their mental state. Hence, these inherent textual features (linguistic (use of nouns, pronouns, etc.) and semantic (use of entity mentions)) can be leveraged by a sequential model for predicting suicide risk [75]. Motivated by prior findings suggests that LSTM selectively filters irrelevant information while maintaining temporal relations; we incorporated them for our Time-variant framework [76, 77]. Consider an example post; "*I dont know man. It is messed up. I dont even go to the exams, but I tell my parents that this time I might pass those exams and will be able to graduate. And parents get super excited and proud of me. It is like Im playing some kind of a Illness joke on my poor family.*" Words like "messed up", "exams", "parents", "graduate", "proud", "illness", "joke", "family" constitute important terms describing the example post. LSTMs learn a representation of a sequence. Our LSTM model predicts the likelihood of each suicidal severity category of a Reddit post, taking into account its sequence of words. However, the representation of a post is learned independently; hence patterns across multiple posts are not recognized. There are ways to improve an LSTM's capability by truncating the sequences, summarizing the post sequences, or random sampling; however, they require human engineering [78].

We require a model which engineers features across multiple posts from a user. Convolutional neural networks (CNNs) are state of the art for such tasks [79]. Thus, we append CNN to learn over LSTM generated probabilities across multiple posts made by a user. Following this intuition, we develop an architecture stacking CNN over LSTM to predict user-level suicidality (see Fig 5).

**TinvM to suicide risk prediction.** Considers learning over all the posts made by a user to provide a user-level suicidality prediction. For this methodology, we put together all the posts made by the user (irrespective of time) in SuicideWatch and other mental-health related

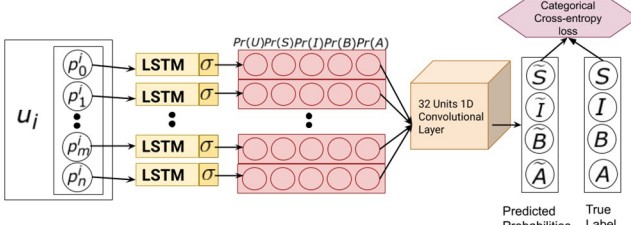

**Fig 5. Time-variant user level suicide risk prediction using LSTM+CNN.** It comprises of an LSTM model to generate probabilities of a post ($p_0^i$), which is a sequence of word embeddings. Inlined CNN model that convolves over a sequence of post-level probabilities ($[Pr(S)Pr(I)Pr(B)Pr(A)]$) to predict user-level ($u_i$) suicide risk.

**Table 3. Example posts from a user($u_i$) and prediction from TinvM.** The italicized text are phrases which contributed to the representation of the post. These phrases had similarity to the concepts in medical knowledge bases.

| |
|---|
| **User Post**: "Homie, . . . Im 27 yo, . . . the job is underpaying—700 euros per month . . . too afraid to search for a new job. . . . fuck me, I guess? . . . had these *thoughts of suicide* and these *fears* to *take charge of my life* from like the end of a high school. 10 years same feelings of dread, same *thoughts of killing myself*." "One day . . .. sudden realization . . . I gonna gather determination . . . roll over the bridge. And my parents, or have a nice heart attack! *feel trapped*. . . . nothing gonna change. You will end up just like me, *roll over the bridge*""No wife, no house, no car, no decent job. Every single day . . . hating myself at work . . .. Im going to *kill myself today* or tomorrow. Probably . . . middle of next week, but the chances are . . . *going to sleep forever*". "I dont even go to the exams . . . I might pass those exams . . . will not graduate . . .. playing some kind of a *Illness joke* . . .my poor family." |
| **User-level Predicted Suicide Risk Severity**: Suicide Behavior |

subreddits. TinvM possesses the capability to learn rich and complex feature representation of the sentences utilizing a deep CNN. Table 3 shows an aggregation of discrete posts of same user in TvarM (Table 4) to predict the C-SSRS suicide risk levels. Our implementation of CNN is well described in Gaur et al. [13] and Kim et al. [80] and is suitable for our contextual classification task [81]. The model takes embeddings of user posts as input and classifies them into one of the suicide risk severity levels. We concatenate embeddings of posts for each user and pass them into the model. Evaluations are performed using the formulations described by Gaur et al. [13].

## Results

We present an evaluation of the two methodologies: TinvM and TvarM, in a cross-validation framework using data from 448 users. We then obtained key insights into throwaway accounts, supportive posts, and uninformative posts. Through an ablation study using different user-types and content-types, we compare TinvM and TvarM models in the user-level suicide risk severity prediction.

### Ablation study

We began our ablation studies with the TinvM setting, as shown in Table 5a. As can be seen from the table, experiment S1, which includes throwaway accounts, uninformative posts, and

**Table 4. Example posts from a user($u_i$) ordered by timestamp (TS) and prediction from TvarM.** The italicized text are phrases which contributed to the representation of each post. These phrases had similarity to the concepts in medical knowledge bases.

| |
|---|
| **Post 1 (TS 1)**: "Homie, . . . Im 27 yo, . . . the *job is underpaying*—700 euros per month . . . too afraid to search for a new job. . . . fuck me, I guess? . . . had these *thoughts of suicide* and these *fears* to *take charge of my life* from like the end of a high school. 10 years same *feelings of dread*, same *thoughts of killing myself*." |
| **Predicted Suicide Risk Severity**: Suicide Ideation |
| **Post 2 (TS 2)**: "One day . . .. sudden realization . . . I gonna gather determination . . . *roll over the bridge*. And my parents, or have a nice *heart attack! feel trapped*. . . . *nothing gonna change*. You will *end up* just like me. I will *roll over the bridge*" |
| **Predicted Suicide Risk Severity**: Suicide Behavior |
| **Post 3 (TS 3)**: "No wife, no house, no car, *no decent job*. Every single day . . . *hating myself* at work . . .. Im going to *kill myself today* or tomorrow. Probably . . . middle of next week, but the chances are . . . *going to sleep forever*" |
| **Predicted Suicide Risk Severity**: Suicide Behavior |
| **Post 4 (TS 4)**: "I dont even go to the *exams* . . . I might pass those exams . . . will *not graduate* . . .. playing some kind of a *Illness joke* . . .my poor family." |
| **Predicted Suicide Risk Severity**: uninformative |
| **User-level Predicted Suicide Risk Severity**: Suicide Ideation |

**Table 5. An ablation study performed on Throwaway accounts (TA; User types), Supportive (SU), and Un-Informative(UI) Posts (Content-types) to evaluate the performance of suicide risk assessment frameworks in Time-invariant (a and b) and Time-variant (c and d) settings.** In the TinvM context, irrespective of user-type, all types of content are required for high precision and high recall in predicting user-level suicidality. Lengthy posts expressing mental health conditions are often made by TA (a), which resulted in high precision compared to Non-TA (b). However, in the TvarM, seldom supportive behavior of suicidal users is important for assessing their suicidality (c). For Non-TA, there is a trade-off between precision and recall concerning uninformative posts. Still, supportive posts help determine the severity of an individual's suicide risk (d). For clinical-grounding based assessment, we recorded the results in Table 7.

| SNo. | TA | UI | SU | Avg. Prec. | Avg. Rec. | F1 | SNo. | TA | UI | SU | Avg. Prec. | Avg. Rec. | F1 |
|---|---|---|---|---|---|---|---|---|---|---|---|---|---|
| S1 | yes | yes | yes | 0.70 | 0.58 | 0.63 | S5 | no | yes | yes | 0.64 | 0.56 | 0.60 |
| S2 | yes | yes | no | 0.58 | 0.48 | 0.52 | S6 | no | yes | no | 0.59 | 0.50 | 0.55 |
| S3 | yes | no | yes | 0.44 | 0.49 | 0.46 | S7 | no | no | yes | 0.45 | 0.42 | 0.43 |
| S4 | yes | no | no | 0.55 | 0.50 | 0.53 | S8 | no | no | no | 0.55 | 0.53 | 0.54 |
| (a) TinvM with Throwaway Accounts | | | | | | | (b) TinvM without Throwaway Accounts | | | | | | |
| S9 | yes | yes | yes | 0.71 | 0.66 | 0.68 | S13 | no | yes | yes | 0.80 | 0.65 | 0.76 |
| S10 | yes | yes | no | 1.0 | 0.45 | 0.62 | S14 | no | yes | no | 1.0 | 0.34 | 0.50 |
| S11 | yes | no | yes | 1.0 | 0.66 | 0.79 | S15 | no | no | yes | 1.0 | 0.63 | 0.77 |
| S12 | yes | no | no | 1.0 | 0.49 | 0.66 | S16 | no | no | no | 1.0 | 0.50 | 0.66 |
| (c) TvarM with Throwaway Accounts | | | | | | | (d) TvarM without Throwaway Accounts | | | | | | |

supportive posts, achieved the best performance. In experiments, S2, S3, and S4, which either exclude uninformative or supportive contents, we observed an average decline of 18% in precision and 9% in the recall. Hence, users' uninformative and supportive contents in r/SuicideWatch were important for suicide risk assessment in TinvM modeling. The modest improvement in precision and recall in suicidality prediction of throwaway accounts is because of verbosity in content compared to non-throwaway accounts. While throwaway accounts have largely been ignored in the previous studies, we noticed useful information on suicidality in their content (see Table 6) [82]. We hypothesize that this is because users are more open to express their emotions and feelings when they can remain anonymous. In another ablation study of TvarM for predicting the suicidality of throwaway accounts, we note a significant decline in false negatives compared to TinvM. We found supportive posts to be more important in determining user-level suicidality (S11 in Table 5c) compared to uninformative posts. This is because contents from a supportive user include past suicidal experiences, which could be higher in suicide severity, causing the TinvM model to predict false positives. The dense content structure of throwaway accounts at each time step improved the averaged recall in experiment S9 (TvarM) compared to S1 (TinvM). Thus, the time-variant modeling is akin to a hypothetical bi-weekly diagnostic interview between a patient and a clinician conducted in a clinical setting. The clinician records per-visit interview with a patient and utilizes

**Table 6. Example informative posts published by throwaway accounts and supportive users on r/SuicideWatch.**

**Throwaway User Post**: "I wanted advice on how to push away any thought of offing myself that might be comforting. I enjoy the quality time I spend with myself, . . .. great friends. We are all alone at times . . . get over it . . ., its not even close to the problem. If you have advice on how to dismiss these sort of thoughts . . .." "Basically, been constantly Delusional disorder . . . the past couple of months, . . . extremely detrimental to my mental health . . . especially bad . . . making it extremely difficult to reach out for help . . . stop talking with one of my good friends The Delusional disorder was so bad . . . situation makes me want to disappear because . . . it makes me feel isolated . . . no chance . . . things will get better."

**Predicted Suicide Risk**: Suicide Ideation (True Label: Suicide Ideation)

**Supportive User's Post**: "I know that you are worthy of many things. I am here to help. I just helped a guy who was thinking of killing himself. Hoping he hasnt and he is okay"

**Predicted Suicide Risk**: Supportive (True Label: Supportive)

**Table 7. Suicide risk severity category-based performance evaluation of TvarM (left) and TinvM (right) approaches.**

| Severity Levels | Avg. Prec. | Avg. Rec. | Severity Levels | Avg. Prec. | Avg. Rec. |
|---|---|---|---|---|---|
| Supportive | 0.64 | 1.0 | Supportive | 0.52 | 1.0 |
| Ideation | 0.83 | 0.65 | Ideation | 0.74 | 0.64 |
| Behavior | 0.78 | 0.23 | Behavior | 0.39 | 0.66 |
| Attempt | 1.0 | 0.41 | Attempt | 1.0 | 0.57 |

it to enhance his/her decision making [83]. Similarly, there is a substantial improvement of 20% and 14% in average precision and recall for non-throwaway accounts in TvarM compared to TinvM (see Table 5b and 5d). The reduction in false positives and false negatives is due to sequence-preserving representations of time-ordered content, capturing local information about suicidality, and keeping important characteristic features across multiple posts through a max-pooled CNN.

## Suicide risk severity category analysis

We characterize the capability of TinvM and TvarM frameworks to predict possible suicide risk severity levels and, subsequently, discuss the functioning of each framework qualitatively.

According to Table 7, for people who are showing signs of suicidal ideation, the time-variant model is better than the time-invariant model (a 5.5% improvement in F1-score). On the other hand, suicidal behavior is better captured by the time-invariant model (a 26.5% improvement in F1-score). Users labeled as suicide attempts did not show variation in the C-SSRS levels of suicide risk over time in their content; hence TinvM performed relatively better than TvarM (a 21% improvement in F1-score). Apart from these categories, there are users on Reddit supporting others by sharing their experiences, who confound suicide risk analysis [13]. We found that TinvM is relatively more susceptible to misclassifying supportive users as suicidal compared to TvarM (a 6.4% improvement in F1-score).

## ROC analysis

The TinvM model was 25% more sensitive at identifying conversations about suicide attempts compared to TvarM. The solid lines representing suicide attempt in ROC curves (see Fig 6) show a significant improvement in recall for TinvM (40% True Positive Rate (TPR) at 20% False Positive Rate (FPR)) compared to TvarM (20% TPR at 20% FPR). However, TinvM had difficulty separating supportive users from suicide attempters, contributing to increases in false negatives (see Table 7). As, users with suicide behavior did not show a significant change in suicide-related terms (repeated use of phrases, such as "loaded gun", "alcoholic parents", "slow poisoning", "scars of abuse"), TinvM model correctly identified 12.5% more users with suicide behaviors compared to TvarM. Further, TinvM predicted 20% of users with suicide behavior as supportive compared to 42% by TvarM, making time-sensitive modeling susceptible to false negatives in high suicide risk severity levels. Users with suicidal ideations show high oscillations in suicidal signals, making TvarM capable of correctly classifying 65% of suicide ideation users, with 20% being misclassified to high severity levels. The false positives occur when users with suicidal ideation use future tense to express suicide behavior. For example, in the following sentence, *For not able to make anything right, getting abused, I would buy a gun and burn my brain*, the user used a future tense to describe ideations, signaling a false positive.

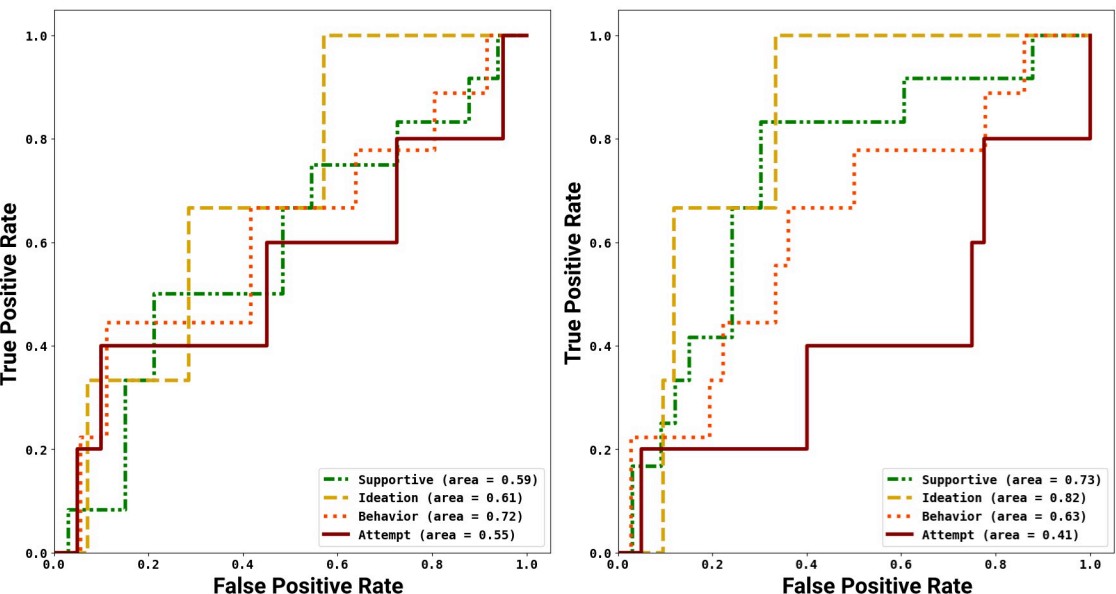

**Fig 6. The ROC plots shows the capability of either approaches in detecting users with different levels of suicide risk severity based on their behavior over time on SW subreddit.** We notice that TvarM (right) is effective in detecting supportive and ideation users. TinvM (left) is capable of detecting behavior and attempt users. We also record that a hybrid of TinvM and TvarM is required for detecting users with suicidal behaviors.

A significant improvement of 26% in AUC for TvarM shows the low sensitivity and high specificity compared to TinvM (TvarM: TPR = 1.0 at FPR = 0.38 compared to TinvM: TPR = 1.0 at FPR = 0.6). Supportive users on MH-Reddit account for the high false positives in the prediction of suicide assessment because of the substantial overlap in the content with users having ideation, behavior, and attempts. The time-variant methodology discreetly identifies semantic and linguistic markers which separate supportive users from users with a high risk of suicide. The representation of words such as "experience", "sharing", "explain", "been there", "help you", past tense in posts, and subordinate conjunctions were preserved in TvarM. However, it is overridden by high-frequency suicide-related terms in TinvM, leading to high false positives [84]. From the ROC curves in Fig 6, TvarM is more specific and less sensitive than TinvM with a 20% improvement in AUC in identifying supportive users. Furthermore, we noticed 80% TPR for TvarM compared to 50% for TinvM at 40% FPR because the expression of helpfulness and care is preserved in TvarM, which are lost due to sudden rise in suicide-related words in TinvM, causing a significant increase in FPR.

## Discussion

Through our quantitative and qualitative experiments, we posit that there is a significant influence of User Types and Content Types on TinvM and TvarM models for suicide-risk assessment. We investigate the influence of throwaway accounts and supportive posts on TinvM and TvarM in estimating the suicide risk of individuals. Reddit users with anonymous throwaway accounts post on stigmatizing topics concerning mental health [85]. Their content is substantial in terms of context and length of posts (Table 6). We highlight the following key takeaways from an ablation study on throwaway accounts: (1) More often, through such accounts, Reddit users seek support as opposed to being support providers. We inspected it on r/SuicideWatch and its associated subreddits (r/depression, r/BPD, r/selfharm, and r/bipolar). In this forum, users often share lengthy posts that directly or indirectly indicate suicide-related

ideations and/or behaviors. (2) TinvM suffers from low recall due to the presence of superfluous content in the posts, but TvarM remains insensitive to uninformative content (see Table 5c). (3) Through time-variant learning over the posts made by throwaway accounts, we noticed a high fluctuation in suicide risk severity levels accounting for low recall in TinvM compared to that in TvarM (seen in Tables 5a amd 5c). Yang et al. describe the modeling of different roles defined for a user in online health communities [86].

Supportive users as defined by our domain experts are either users with their own personal history of suicide-related ideations and behaviors (with or without other mental health disorder comorbidities) or MHPs acting in a professional capacity. Identification of these users types has been of paramount significance in conceptualizing precision and recall (see Table 5a–5d).

We enumerate key insights on the influence of supportive posts on the performance of TinvM and TvarM models: (1) Identifying supportive users in r/SuicideWatch is important to distinguish users with suicide-related ideations from users with suicide-related behavior and/or suicide attempts. From the results in Table 5a–5d, we notice a substantial decline in recall after removing supportive posts (which also removes supportive users). (2) Users with throwaway or non-throwaway accounts can share supportive posts. Removing supporting content makes it harder for the model to distinguish supportive users from non-supportive (suicide ideation, behavior, and attempt) users, causing an increase in false negatives. (3) Both models (TinvM and TvarM) are sensitive to supportive posts (and users); however, TvarM learns representative features that identify supportive users better than TinvM. The aggregation of all helpful posts misclassifies a supportive user as a user with a prior history of suicide-related behaviors or suicide attempts. For example, T0 (timestamps): *I have been in these weird times, having made 30ish cuts on each arm, 60 in total to give myself the pain I gave to my loved ones. I was so much into hurting myself that I plan on killing, but caught hold of hope*; T1: *I realized that I was mean to myself, started looking for therapy sessions, and ways to feel concerned about myself*; TinvM classifies this user as having suicidal behavior, whereas TvarM predicts the user as supportive (true label). This is because phrases such as "feel concerned", "therapy sessions" are prioritized over suicide-related concepts in TinvM, providing correct classification.

An important challenge for TinvM and TvarM models is to distinguish supportive users from non-supportive users on Reddit. From Table 7, we note that the TvarM model has 21% fewer false positives than TinvM. Since content from supportive users semantically overlaps with content from non-supportive users (see Table 8), temporally learning suicide-specific patterns is more effective than learning them in aggregate. Identifying an individual when they are exhibiting suicide-related ideations or behaviors would provide significant benefit in making timely interventions to include creating an effective suicide prevention safety plan. Since prolonged suicide-related ideations and/or suicide-related behaviors are causes of future suicide attempts, they are considered early intervention categories of suicidality.

The methodology which correctly classifies a user either with suicide-related ideations (true positive) or higher suicide risk while maintaining high recall is desired (Table 8c). Based on our series of experiments, we identified TvarM as efficient for the early detection of suicidal ideation or planning compared to the TinvM approach. On F1-score, we recorded a 5.5% improvement with TvarM compared to TinvM. From Table 7, we observe a relatively high recall for TinvM compared to TvarM while detecting users with suicide-related behaviors and suicide attempts.

Reddit posts from either suicide attempters or users with suicide-related behaviors are verbose with words expressing the intention of self-harm. Users may make a chain of posts to explain their mental health status. If these posts are studied in time-slices, the model may not identify the user as suicidal. We see a hybrid model (the composition of TinvM and TvarM)

**Table 8. Qualitative comparison of TinvM and TvarM models representative posts from users who are either supportive or showing signs of suicide ideations, behaviors or attempt.**

| TinvM Pred. | TvarM Pred. | SW Reddit Post or Comments | TinvM Pred. | TvarM Pred. | SW Reddit Post or Comments |
|---|---|---|---|---|---|
| Ideation | Support | "Of many experiences of paranoia, anxiety, guilt, forcing me to jump into a death pithole,. . .. I realized how worthy I m of many things . . . would be giving you my experience on this subreddit" | Behavior | Behavior | "Please listen, I doubt myself and think commiting suicide to escape my situation. Patience, I heard countless times but dying is still a bold decision for me." |
| | Support | "I was a loner, facing increase strokes of anxiety and paranoia, that I went on driving myself into a pithole. I was missing one person who I cared the most . . . .. I feel tired and careless towards anything . . . Guilt of not saving her" | | Attempt | "This may be my last appearance. A thoughtful attempt to take my life is what I left with. I have ordered the materials required for my Suicide this evening. I also have a backup supplier in case my primary source sees through my lies and refuses sale." |
| **(a)** *true label: Support* | | | **(b)** *true label: Behavior* | | |
| Behavior | Ideation | "Thank you. I actually am not on any medication. I was on Zyprexa and then Seroquel for quite a while but stopped taking the anti-psychotics about a year ago." | Attempt | Ideation | "My dad asked to step out of the house. I feared the ugly look and how disgusting I am looking. I tried therapy, talked to strangers. Everday is a torture for me. I like crafts but feel lack of energy in my self" |
| | Ideation | "Anyway, Ive been thinking about seeing my shrink for a while. Maybe get back on the anti-depressants or something. Thank you though for the thoughtful post. It actually means a lot to me since I dont have many friends" | | Ideation | "Dark overwhelming sadness and hyperactive behavior is what describes me. I am trying to live my time to see if something changes for me" |
| **(c)** *true label: Ideation* | | | **(d)** *true label: Attempt* | | |

Pred.: Predictions, SW: r/SuicideWatch.

holding promise for early intervention. (Table 8b). Words like cut, bucket, rope, anesthesia, and methanol are identified by the model as important when all the posts from a user are aggregated; thus, making the time-invariant model appropriate to classify users' attempts into a suicide risk level. From the suicide risk severity based analysis, we infer that time-variant analysis is not applicable across every level of C-SSRS, and time-invariant modeling is required to estimate the risk better. Further, from human behavior shown in the online conversations, users do not express signals of suicide attempts or suicide-related behavior in all their posts. Hence, TvarM learning fails to capture the nuances identified by TinvM (see Table 8d).

## Clinical relevance of the study

Psychiatrists treat a siloed community of patients suffering from mental health disorders, which restricts diversification. Information obtained from EHRs provide time-bounded individual-level insights throughout multiple appointments, often as infrequent as every 3-6 months. For the MHP, patient monitoring in this traditional structure was, until recently, the only pragmatic option outside of resource-consuming intra-appointment telephone calls or community-partnered wellness visits. The value of closer-to-real-time patient status updates is realized when specific and appropriate markers activate timely interventions prior to an impending suicide-related behavior or attempt. As compared to prior studies attempting to make similar identifications using Reddit user posts, this work uniquely combines state-of-the-art deep learning models with long-standing, clinically-common metrics of suicide risk. Notably, Reddit communications were treated to special translation into the domain of clinically-established tools and nomenclature to reveal the signal of suicide-related ideations and behaviors—otherwise hopelessly buried under more information than could be sorted by even the most diligent MHP. Decades of research has supported the notion that early targeted interventions are critical for reducing suicide rates [87–89].

Consistent with PLoS One Open Data recommendation [90], we are making this data available via zenodo.org [91]

## Limitations and future work

The result of this study is to develop an expert-in-the-loop technology for web-based intervention for a terminal mental health condition, suicide, and perform technology evaluation from the perspective of explainability [32, 92]. Through concrete outcomes defining the capability of TinvM and TvarM, it is evident a hybrid model is desired to estimate the likelihood of an individual to exhibit suicide-related ideations, suicide-related behaviors, or suicide attempts for precise intervention. However, there are certain limitations of our current research that also motivates future work. First, both TinvM and TvarM were able to track the changing nature of concepts but ignore causal relationships. For instance, "which mental health condition, symptom, side-effects, or medication made the user drift from one community to another." Second, our study excludes any support given to the support seeking an individual in the form of comments. Third, we assume an individual's static role seeking help to cope with suicide-related ideations or suicide-related behaviors. This ignores any change in behavior that would make the same individual helpful/supportive to others. Fourth, the number of labeled instances for training covers users and posts from 8 out of 15 (r/bipolar, r/BPD, r/depression, r/anxiety, r/opiates, r/OpiatesRecovery, r/selfharm, r/StopSelfHarm, r/BipolarSOs, r/addiction, r/schizophrenia, r/autism, r/aspergers, r/cripplingalcoholism, and r/BipolarReddit) (54%) mental health subreddits. Indeed, the data is not representing individuals suffering from other mental health conditions or comorbidity. Apart from the domain-specific limitations in our current research, we noticed the problem of handling intentional and unintentional biases to be unaddressed. The intentional bias in the form of knowledge (also called Knowledge bias) is augmented through contextualization and abstraction. Knowledge bias is a necessary evil as it helps explain pattern recognition, but its quantification based on the model's (TinvM and TvarM) need has not been explored. The unintentional bias in the form of aggregation occurring in TinvM assumes no significant change in an individual's suicide-related behaviors over time. Another form of aggregation bias in both TinvM and TvarM is using the concatenation function to represent a post or a user. Though such operations are problematic when not guided by knowledge, it would hurt the outcome's explainability, but it is not the case in our current research.

## Supporting information

**S1 Dataset.**
(TXT)

## Author Contributions

**Conceptualization:** Manas Gaur.

**Data curation:** Vamsi Aribandi, Amanuel Alambo, Ugur Kursuncu, Jonathan Beich.

**Formal analysis:** Manas Gaur, Krishnaprasad Thirunarayan.

**Funding acquisition:** Jyotishman Pathak, Amit Sheth.

**Methodology:** Manas Gaur, Vamsi Aribandi, Amanuel Alambo, Krishnaprasad Thirunarayan.

**Project administration:** Manas Gaur, Ugur Kursuncu.

**Resources:** Manas Gaur.

**Supervision:** Jyotishman Pathak, Amit Sheth.

**Validation:** Amanuel Alambo, Ugur Kursuncu, Jonathan Beich.

**Visualization:** Manas Gaur, Vamsi Aribandi.

**Writing – original draft:** Manas Gaur, Vamsi Aribandi, Krishnaprasad Thirunarayan.

**Writing – review & editing:** Manas Gaur, Amanuel Alambo, Ugur Kursuncu, Krishnaprasad Thirunarayan, Jyotishman Pathak, Amit Sheth.

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
