## [Decision Letter · Decision Letter 0]

29 Dec 2020

PONE-D-20-23705

Characterization of Time-variant and Time-invariant Assessment of Suicidality on Reddit using C-SSRS

PLOS ONE

Dear Dr. Sheth,

Thank you for submitting your manuscript to PLOS ONE. After careful consideration, we feel that it has merit but does not fully meet PLOS ONE’s publication criteria as it currently stands. Therefore, we invite you to submit a revised version of the manuscript that addresses the points raised during the review process.

We look forward to receiving your revised manuscript.

Kind regards,

Vincenzo De Luca

Academic Editor

PLOS ONE

Journal Requirements:

Reviewers' comments:

Reviewer's Responses to Questions

**Comments to the Author**

1. Is the manuscript technically sound, and do the data support the conclusions?

Reviewer #1: Yes

Reviewer #2: Yes

2. Has the statistical analysis been performed appropriately and rigorously? 

Reviewer #1: Yes

Reviewer #2: Yes

3. Have the authors made all data underlying the findings in their manuscript fully available?

Reviewer #1: Yes

Reviewer #2: Yes

4. Is the manuscript presented in an intelligible fashion and written in standard English?

Reviewer #1: Yes

Reviewer #2: Yes

5. Review Comments to the Author

Reviewer #1: Novel study addressing the knowledge gap pertaining to Time-variant and Time-invariant assessment of suicidality. The manuscript is written in comprehensible form and present possible algorithms to assess suicide risk.

Abstract, Introduction, Materials and methods, results : Well written and presented

Discussion: Support discussion with few more research studies.

References: Volume, Issue and page numbers need to be added

Check for plagiarism

Reviewer #2: The purpose of this study is to compare time-variant and time-invariant modeling to predict suicidal risk among Reddit platform users. The time-variant model better identifies supportive users and those with suicidal ideation, while the time-invariant model allows for better identification of suicidal behaviors and attempts.

This manuscript addresses an important topic and generates interesting results.

General comments:

1. The introduction and the Materials and Methods sections could be more clearly structured. Specific suggestions are made below.

2. In the literature on suicide, there is a lack of consistency in the terms used to report on suicidal thoughts and behaviors. In this manuscript, several different terms are used, and the reporting of suicidal thoughts and behaviors should be standardized throughout the manuscript.

For example, terms such as "successfully die by suicide" (abstract) or "committing suicide" (line 56, p.3) should be avoided and replaced with "death by suicide”. The authors should decide if they use the expression “suicidal thoughts and behaviors” or “suicidality”, or “suicidability” (line 45, p.3).

“Suicidal thoughts and behaviors” (STBs) is a broad term that covers all the behaviors reported in the manuscript and could be used to refer to several STBs. Where it is necessary, the type of STB could be specified (e.g., suicidal ideation, suicidal attempt, death by suicide).

Please refer to these papers for more details:

Silverman MM, Berman AL, Sanddal ND, O'Carroll P W, Joiner TE (2007) Rebuilding the tower of Babel: a revised nomenclature for the study of suicide and suicidal behaviors. Part 2: Suicide-related ideations, communications, and behaviors. Suicide Life Threat Behav 37 (3):264-277. doi:10.1521/suli.2007.37.3.264

O'Carroll PW, Berman AL, Maris RW, Moscicki EK, Tanney BL, Silverman MM (1996) Beyond the Tower of Babel: A Nomenclature for Suicidology. Suicide and Life-Threatening Behavior 26 (3):237-252. doi:10.1111/j.1943-278X.1996.tb00609.x

Abstract:

The abstract captures well the different sections of the manuscript.

3. The following sentence "Stratifying risk in terms of severity and temporality is important in the risk management of a suicidal patient (e.g., involuntary hospitalization)" refers to the clinical relevance of this study and it would be very relevant to address this aspect further in the introduction section.

4. As previously mentioned, the term “successfully die by suicide” should be replaced by “die by suicide”.

Author summary:

5. The term “terminal mental illness” should be replaced. E.g., "suicide attempt survivors" or “individuals who have recovered from severe mental illness” could be used.

6. The statement below could also be further discussed in the manuscript: First, in the Introduction to reinforce the rationale of the study; second, in the Discussion to highlight the relevance of the results.

“Our study can be used to build tools that identify the suicide risk from historical mental health records (e.g., psychotherapy notes) and social media data, and help policymakers and clinical psychiatrists make informed decisions”.

Introduction:

7. Overall, the introduction could be more concise and the clinical relevance of the development of learning algorithms to assess suicide risk could be further discussed.

8. To better contextualize the importance of using time-variant models, the limitations of time-invariant models could be discussed in more details. For example, it is reported that "this approach is limited due to poor patient engagement and treatment adherence » (line 18). The authors could clarify how poor patient engagement or treatment adherence impact time-invariant models.

9. The following sentences could be included in the Discussion section when the findings are discussed rather than in the Introduction:

“Our experiments demonstrate the utility of MH-Reddit as an auxiliary source for patient-level and community-level insights into mental health disorders and suicide risk. Our study could be integrated with clinical diagnostic interviews to reinforce the decisions of MHPs” (line 139-141, p.4).

10. The last paragraph of the introduction (line 143, p.5) should be included in the Materials and Methods section. It would be clearer if the information on the description of the dataset was grouped together.

11. The aim of the study should be explicitly reported at the end of the introduction.

Materials and Methods:

I can’t comment on the dataset creation and the algorithms development since this is not my field of expertise. However, some aspects of the method could be clarified.

12. At the end of the Introduction section (line 143, p.5), there is a dataset of 448 redditors described. Then, the first paragraph of the Material and Method section (p.5) describe a dataset from >550K users, with a focus on users in the SuicideWatch subreddit. It is not clear whether the created dataset of 448 suicidal redditors is drawn from the dataset described in the first paragraph of the Materials and Methods section. This should be clarified.

13. Moreover, it is mentioned at line 144 (p.5) that there is ">98,000 users' content on r/SuicideWatch". Then, line 170 (p.5) indicates "we primarily focused on users in the SuicideWatch subreddit (95K users)". What explains the distinct number of users?

The date (or at least the year) on which the number of users was observed should be mentioned since the number of users in the SuicideWatch subreddit seems to be increasing.

14. The article by Gaur et al, 2019 is often cited for more information on the method to create the dataset. The description in the article by Gaur et al., 2019 seems clearer. The description provided in this manuscript could be revised to include more details from the Gaur et al. 2019 paper to better understand the steps conducted. For example, Figure 3 in Gaur et al. paper is very helpful.

15. In the section Annotator Agreement, it is first reported that a dataset of 500 users was used (line 238, p.7) with reference to the dataset described above. Then it is mentioned that « the annotated dataset of 448 users..." (line 248, p.7). Once the dataset is described at the beginning of the method, as suggested, and Figure 2 is presented, it would avoid confusion to always refer to the 448 users that are included in the study.

Fig 2 : Very helpful figure

Results:

16. Hypotheses, such as the one announced at line 309 (p.9), should be in the Discussion.

17. Interpretation of the results should also be in the Discussion (e.g., line 333-336, p.10)

The post examples are really useful to better understand the results.

Discussion:

18. The study produces interesting results. The discussion could further highlight the clinical implications and implications for future research. More references should be added to the discussion.

19. Strengths and limitations of the study should be presented in the Discussion.

20. It would be interesting to further contrast the results with other studies in the field and make hypotheses to explain the results (such as the hypothesis included in the results section, line 309, p.9).

21. The last sentence of the Discussion (line 455, p.15) should be reworded if it is maintained for the published version of the article.

Minor comments:

22. Cites should be before the period of each sentence (e.g., “[…] mental healthcare providers (MHPs) and patients [1, 4]. Rather than “[…] mental healthcare providers (MHPs) and patients. [1, 4]”

23. Line 48, p.3: The acronym "MH-reddit" should be defined when first used in this sentence. Alternatively, the sentence could simply be reworded "There are three User-Types in mental health subreddits (MH-Reddits)".

24. CNN should be defined the first time it is used (line 280, p.8 rather than line 326, p.9).

25. There seems to be an error between the figures’ numbers announced in the text and those in the appendix. E.g., Figure 3 in the text seems to refer to the fourth figure in the appendix. Figure 4 in the text refers to the fifth figure in the appendix.

26. Tables should be cited in order (e.g., Table 4 should be cited before table 5a).

6. PLOS authors have the option to publish the peer review history of their article (what does this mean?). If published, this will include your full peer review and any attached files.

Reviewer #1: No

Reviewer #2: No

---

## [Author Response · Author response to Decision Letter 0]

18 Feb 2021

We thank the reviewers for their constructive feedback that greatly improved this manuscript. We have summarized the changes we made below based on these feedback.

1. The introduction and the Materials and Methods sections could be more clearly structured. Specific suggestions are made below.

Response: We appreciate this particular comment from the reviewers. We have addressed the points highlighted, specifically for introduction and the Materials and Methods sections, on clarity, clinical relevance, and broader impact of the study. We have made necessary changes in blue color (a color coding followed for track changes). 

2. In the literature on suicide, there is a lack of consistency in the terms used to report on suicidal thoughts and behaviors. In this manuscript, several different terms are used, and the reporting of suicidal thoughts and behaviors should be standardized throughout the manuscript.

Response: We have noticed the lack of such consistency that was brought up by the reviewers providing examples. With our resident psychiatrist co-author, we have carefully reviewed the following two papers for consistency on the terminology related to suicide risk: (a) Silverman MM, Berman AL, Sanddal ND, O'Carroll P W, Joiner TE (2007) Rebuilding the tower of Babel: a revised nomenclature for the study of suicide and suicidal behaviors. (b) O'Carroll PW, Berman AL, Maris RW, Moscicki EK, Tanney BL, Silverman MM (1996) Beyond the Tower of Babel: A Nomenclature for Suicidology. 

Throughout the manuscript, we are using the following standardized terms; “suicide-related ideations, suicide-related behaviors, suicide attempt, and suicide risk assessment” as advocated by the reviewer and the current literature. 

3. The following sentence "Stratifying risk in terms of severity and temporality is important in the risk management of a suicidal patient (e.g., involuntary hospitalization)" refers to the clinical relevance of this study and it would be very relevant to address this aspect further in the introduction section.

Response: We have addressed this aspect in multiple places. Please see the following lines. 18-20, 30-35, 40-44, 104-106, and 151-165. 

4. As previously mentioned, the term “successfully die by suicide” should be replaced by “die by suicide”.

Response: We have replaced the term with following: “complete suicide” (Please see abstract, line no. 2)

5. The term “terminal mental illness” should be replaced. E.g., "suicide attempt survivors" or “individuals who have recovered from severe mental illness” could be used.

Response: For consistency with the terminology, we have used “suicide-related ideation, suicide-related behavior, and suicide attempt”. In addition, we have clarified the initial two paragraphs of the author summary and made the necessary edits in the revised version. 

6. The statement below could also be further discussed in the manuscript: First, in the Introduction to reinforce the rationale of the study; second, in the Discussion to highlight the relevance of the results.

“Our study can be used to build tools that identify the suicide risk from historical mental health records (e.g., psychotherapy notes) and social media data, and help policymakers and clinical psychiatrists make informed decisions”.

Response: We have made changes that reinforces the above statement and explained its relevance to end-users, who are mental healthcare providers in our study. Please refer to paragraph from line no. 151-165. 

7. Overall, the introduction could be more concise and the clinical relevance of the development of learning algorithms to assess suicide risk could be further discussed.

Response: We have made significant efforts in tightening the introduction by removing redundancy and focussing more on the clinical relevance of the study. The introduction also contains the prior research related to this study. 

8. To better contextualize the importance of using time-variant models, the limitations of time-invariant models could be discussed in more details. For example, it is reported that "this approach is limited due to poor patient engagement and treatment adherence » (line 18). The authors could clarify how poor patient engagement or treatment adherence impact time-invariant models.

Response: We have revised the respective statements by mentioning “how poor patient engagement or treatment” impact time-invariant models (see Lines 27-29). Further, we have added a new paragraph on “Limitations” of this study. Please see Lines. 493-522. 

9. The following sentences could be included in the Discussion section when the findings are discussed rather than in the Introduction:

“Our experiments demonstrate the utility of MH-Reddit as an auxiliary source for patient-level and community-level insights into mental health disorders and suicide risk. Our study could be integrated with clinical diagnostic interviews to reinforce the decisions of MHPs” (line 139-141, p.4).

Response: We have moved this paragraph from introduction to discussion (Please see Lines 486-488). 

10. The last paragraph of the introduction (line 143, p.5) should be included in the Materials and Methods section. It would be clearer if the information on the description of the dataset was grouped together.

Response: We moved all the parts of the dataset description into Materials and Methods section. 

11. The aim of the study should be explicitly reported at the end of the introduction.

Response: We explicitly created a paragraph at the end of the introduction that describes the key contributions of this study. Please see Lines 166-175. 

12. At the end of the Introduction section (line 143, p.5), there is a dataset of 448 redditors described. Then, the first paragraph of the Material and Method section (p.5) describes a dataset from >550K users, with a focus on users in the SuicideWatch subreddit. It is not clear whether the created dataset of 448 suicidal redditors is drawn from the dataset described in the first paragraph of the Materials and Methods section. This should be clarified.

Response: We have revised the description of redditors for clarity. Please see the Lines 166-175 (in introduction) and 177-179 in Materials and Methods. Also, we have included Figure 3 (taken from our previous work Gaur et al. 2019) for illustration of the process for dataset creation. 

13. Moreover, it is mentioned at line 144 (p.5) that there is ">98,000 users' content on r/SuicideWatch". Then, line 170 (p.5) indicates "we primarily focused on users in the SuicideWatch subreddit (95K users)". What explains the distinct number of users?

The date (or at least the year) on which the number of users was observed should be mentioned since the number of users in the SuicideWatch subreddit seems to be increasing.

Response: We have updated the number of users in our dataset in the Introduction (Line no. 135) and Materials and Method (line no. 177-179) sections. We have also added Table 1 on Data Statistics. 

14. The article by Gaur et al, 2019 is often cited for more information on the method to create the dataset. The description in the article by Gaur et al., 2019 seems clearer. The description provided in this manuscript could be revised to include more details from the Gaur et al. 2019 paper to better understand the steps conducted. For example, Figure 3 in Gaur et al. paper is very helpful.

Response: We agree that this will provide clarity on the dataset creation. We have included additional summary information and provided reference to the sections in Gaur et al. 2019 for further details. 

15. In the section Annotator Agreement, it is first reported that a dataset of 500 users was used (line 238, p.7) with reference to the dataset described above. Then it is mentioned that « the annotated dataset of 448 users..." (line 248, p.7). Once the dataset is described at the beginning of the method, as suggested, and Figure 2 is presented, it would avoid confusion to always refer to the 448 users that are included in the study.

Response: We have provided clarification on how the dataset of 448 users has been created. Furthermore, Table 1 and Figure 3 now provides more details on the dataset and its creation process. 

16. Hypotheses, such as the one announced at line 309 (p.9), should be in the Discussion.

Response: We have moved the hypothesis to discussion, as suggested by reviewers. Also, we have specifically mentioned relative novelty in this research compared to prior research on suicide and social media (Please see lines. 482-486).

17. Interpretation of the results should also be in the Discussion (e.g., line 333-336, p.10). The post examples are really useful to better understand the results.

Response: We have moved the suggested and similar statements that convey the interpretation of the results to the Discussion section. 

18. The study produces interesting results. The discussion could further highlight the clinical implications and implications for future research. More references should be added to the discussion. 

Response: We have now improved the clinical relevance of study with strength of the research, added a paragraph on limitations and future directions. 

19. Strengths and limitations of the study should be presented in the Discussion.

Response: We have added Limitations of the study in the Discussion section (Please see the line no. 493-522) The strengths of the study have been described in “Clinical relevance of the study” within the Discussion section (Please see the lines 464-491).

20. It would be interesting to further contrast the results with other studies in the field and make hypotheses to explain the results (such as the hypothesis included in the results section, line 309, p.9).

Response: We have moved the hypothesis to discussion, as suggested by reviewers. Also, we have specifically mentioned relative novelty in this research in comparison with prior research on suicide and social media (Please see line no. 482-486).

21. The last sentence of the Discussion (line 455, p.15) should be reworded if it is maintained for the published version of the article.

Response: We have replaced the referred sentence above with the following: “Consistent with PLoS One Open Data recommendation (https://plos.org/open-science/open-data/) , we are making this data available via zenodo.org

(https://zenodo.org/record/2667859#.YCbH6h1OlQI)” (Please see the lines 488-491). 

22. Cites should be before the period of each sentence (e.g., “[…] mental healthcare providers (MHPs) and patients [1, 4]. Rather than “[…] mental healthcare providers (MHPs) and patients. [1, 4]”

Response: All the citations have been made before the period. 

23. Line 48, p.3: The acronym "MH-reddit" should be defined when first used in this sentence. Alternatively, the sentence could simply be reworded "There are three User-Types in mental health subreddits (MH-Reddits)".

Response: We have made the required changes in Line No. 63 and used in subsequent mentions. 

24. CNN should be defined the first time it is used (line 280, p.8 rather than line 326, p.9).

Response: We have made the change in line no. 293 (the first time mention of convolutional neural networks (CNNs))

25. There seems to be an error between the figures’ numbers announced in the text and those in the appendix. E.g., Figure 3 in the text seems to refer to the fourth figure in the appendix. Figure 4 in the text refers to the fifth figure in the appendix.

Response: We have renamed the figures and re-uploaded it on the submission portal. 

26. Tables should be cited in order (e.g., Table 4 should be cited before table 5a).

Response: We have revised the order of Tables. Though there will be a change in the number as a new table has been added (Table 1) on data statistics.

---

## [Decision Letter · Decision Letter 1]

15 Mar 2021

PONE-D-20-23705R1

Characterization of Time-variant and Time-invariant Assessment of Suicidality on Reddit using C-SSRS

PLOS ONE

Dear Dr. Sheth,

Thank you for submitting your manuscript to PLOS ONE. After careful consideration, we feel that it has merit but does not fully meet PLOS ONE’s publication criteria as it currently stands. Therefore, we invite you to submit a revised version of the manuscript that addresses the points raised during the review process.

Please be advised that submitting a revision does not guarantee acceptance.

We look forward to receiving your revised manuscript.

Kind regards,

Vincenzo De Luca

Academic Editor

PLOS ONE

Journal Requirements:

Reviewers' comments:

Reviewer's Responses to Questions

**Comments to the Author**

1. If the authors have adequately addressed your comments raised in a previous round of review and you feel that this manuscript is now acceptable for publication, you may indicate that here to bypass the “Comments to the Author” section, enter your conflict of interest statement in the “Confidential to Editor” section, and submit your "Accept" recommendation.

Reviewer #1: All comments have been addressed

Reviewer #2: All comments have been addressed

2. Is the manuscript technically sound, and do the data support the conclusions?

Reviewer #1: Yes

Reviewer #2: Yes

3. Has the statistical analysis been performed appropriately and rigorously? 

Reviewer #1: Yes

Reviewer #2: Yes

4. Have the authors made all data underlying the findings in their manuscript fully available?

Reviewer #1: Yes

Reviewer #2: Yes

5. Is the manuscript presented in an intelligible fashion and written in standard English?

Reviewer #1: Yes

Reviewer #2: Yes

6. Review Comments to the Author

Reviewer #1: All the reviewer comments have been incorporated in the manuscript by authors. The introduction, Materials and Methods and discussion sections are clearly structured.

Reviewer #2: The authors are very responsive to comments made and have done a good job of improving the rationale of their study. I still have a few minor comments on the revised version of the manuscript.

1. Suicide-related terms are reported much more consistently.However, as mentioned in the O'Carroll paper, the revised term for 'completed suicide' is simply 'suicide' or 'died by suicide'. Efforts are now being made to avoid all terms with "positive" connotations such as "completed" when referring to death by suicide.

2. I am still not sure why the paragraph from line 131 to 150 is not in the method section. This paragraph seems redundant with the paragraph from line 184 to 209 (which is in the method section).

3. Line 177: the number of posts is not indicated, it is written "XX posts”.

4. It would be helpful to define the category "suicide indicator" when it is mentioned that this category is excluded from the study (line 253).

The authors have done a good job of improving the discussion and incorporating the comments made.

7. PLOS authors have the option to publish the peer review history of their article (what does this mean?). If published, this will include your full peer review and any attached files.

Reviewer #1: No

Reviewer #2: No

---

## [Author Response · Author response to Decision Letter 1]

23 Mar 2021

Characterization of Time-variant and Time-invariant Assessment of Suicidality on Reddit using C-SSRS (Summary of Changes)

We thank the reviewers again for their constructive feedback that has greatly improved the quality of this manuscript. We have addressed the changes suggested by the reviewers. Places where we have made edits, as suggested by reviewers, are colored in brown. 

Q1: Suicide-related terms are reported much more consistently.However, as mentioned in the O'Carroll paper, the revised term for 'completed suicide' is simply 'suicide' or 'died by suicide'. Efforts are now being made to avoid all terms with "positive" connotations such as "completed" when referring to death by suicide.

Response: We have made the change in the abstract. 

Q2: I am still not sure why the paragraph from line 131 to 150 is not in the method section. This paragraph seems redundant with the paragraph from line 184 to 209 (which is in the method section).

Response: We thank the reviewer for suggesting the change. We have moved the content from line numbers starting from 131 to 150 in the Introduction section to line numbers 173 to 188 in the Materials and Methods section. 

Q3: Line 177: the number of posts is not indicated, it is written "XX posts”.

Response: We have entered the number of posts in the dataset. 

Q4: It would be helpful to define the category "suicide indicator" when it is mentioned that this category is excluded from the study (line 253).

Response: We have provided the definition for “Suicide Indication” in line numbers starting from 252 to 260.

---

## [Decision Letter · Decision Letter 2]

7 Apr 2021

Characterization of Time-variant and Time-invariant Assessment of Suicidality on Reddit using C-SSRS

PONE-D-20-23705R2

Dear Dr. Sheth,

We’re pleased to inform you that your manuscript has been judged scientifically suitable for publication and will be formally accepted for publication once it meets all outstanding technical requirements.

Kind regards,

Vincenzo De Luca

Academic Editor

PLOS ONE

Additional Editor Comments (optional):

Reviewers' comments:

Reviewer's Responses to Questions

**Comments to the Author**

1. If the authors have adequately addressed your comments raised in a previous round of review and you feel that this manuscript is now acceptable for publication, you may indicate that here to bypass the “Comments to the Author” section, enter your conflict of interest statement in the “Confidential to Editor” section, and submit your "Accept" recommendation.

Reviewer #1: All comments have been addressed

Reviewer #2: All comments have been addressed

2. Is the manuscript technically sound, and do the data support the conclusions?

Reviewer #1: Yes

Reviewer #2: Yes

3. Has the statistical analysis been performed appropriately and rigorously? 

Reviewer #1: Yes

Reviewer #2: Yes

4. Have the authors made all data underlying the findings in their manuscript fully available?

Reviewer #1: Yes

Reviewer #2: Yes

5. Is the manuscript presented in an intelligible fashion and written in standard English?

Reviewer #1: Yes

Reviewer #2: Yes

6. Review Comments to the Author

Reviewer #1: Authors have adequately addressed comments raised in a previous round of review. All sections of manuscript are presented in intelligible way.

Reviewer #2: The authors have incorporated all the comments that were made. I have no additional comments.

7. PLOS authors have the option to publish the peer review history of their article (what does this mean?). If published, this will include your full peer review and any attached files.

Reviewer #1: No

Reviewer #2: No

---

## [Editor Report · Acceptance letter]

4 May 2021

PONE-D-20-23705R2 

Characterization of Time-variant and Time-invariant Assessment of Suicidality on Reddit using C-SSRS  

Dear Dr. Sheth:

I'm pleased to inform you that your manuscript has been deemed suitable for publication in PLOS ONE. Congratulations! Your manuscript is now with our production department. 

Kind regards, 

on behalf of

Dr. Vincenzo De Luca 

Academic Editor

PLOS ONE